# Phenolic Compounds in Fractionated Blackcurrant Leaf Extracts in Relation to the Biological Activity of the Extracts

**DOI:** 10.3390/molecules28227459

**Published:** 2023-11-07

**Authors:** Monika Staszowska-Karkut, Barbara Chilczuk, Małgorzata Materska, Renata Kontek, Beata Marciniak

**Affiliations:** 1Department of Chemistry, Faculty of Food Science and Biotechnology, University of Life Sciences in Lublin, Akademicka 15, 20-950 Lublin, Poland; monika.staszowska-karkut@up.lublin.pl (M.S.-K.); malgorzata.materska@up.lublin.pl (M.M.); 2Department of Molecular Biotechnology and Genetics, Faculty of Biology and Environmental Protection, University of Lodz, Banacha 12/16, 90-237 Lodz, Poland; renata.kontek@biol.uni.lodz.pl (R.K.); beata.marciniak@biol.uni.lodz.pl (B.M.)

**Keywords:** blackcurrant leaf, *Ribes nigrum* L., phenolic compounds, anticancer activity, antiradical activity

## Abstract

The aim of this study was to determine the relationship between antioxidant and anticancer properties of extracts from blackcurrant (*Ribes nigrum* L.) leaves and their fractions and chemical contents. Dried ethanolic extract was divided into three fractions using solid phase extraction: aqueous (F1), 40% MeOH (F2), and 70% MeOH (F3). Both the extract and the fractions were analyzed in terms of antiradical activity (DPPH^•^ and ABTS^+•^), total phenolic compounds, and total flavonoids. The antitumor potential of the fractions was evaluated in vitro on human colorectal (HCT 116) and prostate (PC-3) cancer cells. Phenolics were identified using HPLC-QTOF-MS, and twelve compounds were quantified by HPLC-DAD. Finally, principal component analysis was carried out to assess the relationship between the tested factors. The results confirmed that blackcurrant leaves are a rich source of phenolics with high antioxidant activity and anticancer properties. It was demonstrated that the F2 fraction had the highest content of phenolics and the highest antiradical activity. Additionally, only this fraction showed cytotoxic activity against HCT 116 cells. It was confirmed that both the blackcurrant leaf extract and its fractions are a promising source of condensed active compounds and can be used as natural functional food additives.

## 1. Introduction

The growing public awareness of the principles of nutrition forces producers to pay more attention to the quality of food being produced. Nutrients supplied with meals are not only building blocks and a source of energy for the body but also components of all proteins and enzymes determining proper functions of the organism. Therefore, the quality of food and a balanced diet largely determine homeostasis in the human body and health. Many available studies confirm the health benefits of consumption of low-processed foods [1]. This has increased the interest in the health-promoting properties of fruits, vegetables, and herbs. In addition, consumers are increasingly willing to use preparations and medicines based on natural ingredients as well as natural methods of treatment and enhancement of their immunity. Hence, plant extracts and decoctions, which are convenient and relatively durable, are often used [2]. The biological properties of plant extracts are determined by their chemical composition. Phenolic acids and flavonoids present therein have antioxidant, antibacterial, antiseptic, and hemostatic effects [2,3]. These compounds also exhibit anticancer properties by blocking the synthesis of carcinogenic substances formed through the metabolic transformation of carcinogenic compounds [4].

Blackcurrant (*Ribes nigrum* L.) fruits are a popular plant species, often consumed as raw or processed products [5,6]. They are a rich source of vitamin C and flavonoids: catechins, epicatechins, procyanidins, kaempferol, and myricetin [6]. Numerous literature data confirm their high biological activity, e.g., the inhibition of cell proliferation and antimutagenic, antibacterial, anti-inflammatory, antitumor, and antihypertensive properties [7,8,9,10]. In addition to the fruits, the leaves of this plant are also a valuable raw material containing many phenolic compounds whose concentrations and types are different than the fruit’s [11]. Leaves harvested at maturity, when they reach a typical size for the species, are characterized by the highest content of phenolic compounds dominated by quercetin and its derivatives as well as myricetin [12,13]. The unique fatty acids contained in blackcurrant leaves are α-linolenic acid, cis-7, 10, 13-hexadecatrienoic acid (16:3), stearidine acid (18:4), and γ-linolenic acid (γ-18:3). These acids exhibit anticancer, anti-inflammatory, and antimicrobial properties [14].

Cancer is currently one of the most common causes of death in the world; hence, research conducted in many scientific centers focuses on finding effective ways to prevent and treat neoplastic diseases. The search for natural compounds that can be used as an adjuvant in the treatment of certain cancers is important. Blackcurrant fruits have been studied in this respect, and many studies indicate their beneficial effect in the prevention of cancer cell development. For example, fruit extract was shown to inhibit the growth of HT-29/HCT 116 colon cancer cells, MCF-7 breast cancer cells, and human HL-60 promyelocytic leukemia cells [7,9]. Jia et al. [15] investigated the free radical activity of blackcurrant extract and the proliferation of SGC-7901 gastric cancer cells. They proved that the extract inhibited the proliferation of the analyzed SGC-7901 cells in a dose- and time-dependent manner. The flow cytometry analysis carried out by these authors showed that treatment at concentrations of 10–20 mg/mL resulted in a significant reduction in the number of viable cells. The authors found that the phenolic compounds present in the extract at high concentrations (12.2 mg/mL), including six anthocyanins identified by HPLC-ESI-MS2, appeared to be responsible for the antiradical and antitumor activity of the extract [15]. In contrast to the fruits, blackcurrant leaves have been analyzed much less frequently in terms of their use as a source of bioactive compounds. Their potential as myeloperoxidase (MPO) inhibitors [10] and activators of nitric oxide synthase (eNOS) [16] as well as their antibacterial and anti-inflammatory activity [17,18,19,20] have been reported. In their study of the activity of extracts of leaves from four fruiting shrubs, including blackcurrant leaves, Ziemlewska et al. [21] showed the potential of these raw materials to be used in cosmetology as an ingredient in anti-aging cosmetics. However, the anticancer properties of leaf extracts have been studied sporadically. Only Paunovic and Maskovic [17] analyzed three cell lines, human rhabdomyosarcoma (RD), human cervix carcinoma Hep2c (HeLa), and a line derived from murine fibroblasts (L2OB), and showed antitumor activity of blackcurrant leaf extracts, which was lower than that of fruit extracts.

Since blackcurrant leaves are an easily available waste raw material that can be used in the food, cosmetics, and medicine industries, we decided to investigate their biological properties, including antioxidant and anticancer properties. In addition, our research focused not only on the extract but also on its fractions. The extract was divided into three groups of compounds with similar lipophilic properties. This contributed to the accumulation of active compounds of a similar nature, which modulated the properties of the fractions and greatly facilitated chromatographic analysis.

The aim of this research was to determine the relationship between the chemical composition of fractionated extracts and their biological activity and to prove that fractionation is a good method for obtaining plant preparations intended for versatile use.

## 2. Results and Discussion

### 2.1. Phenolic Content and Antioxidant Activity

The fractionation of the crude ethanolic blackcurrant leaf extract separated fractions with increasing lipophilicity. The yield of this process depended on the nature of the isolated compounds and was highest for fraction F1—36.8%, F2—21.2%, and F3—3.9% per 1 g of dry extract. Among the analyzed fractions, the F2 fraction eluted with a 40% solution of methanol in water turned out to have the highest amounts of active compounds (Table 1). It contained 23% higher content of phenolic compounds expressed as gallic acid equivalents than the ethanol extract and as much as 87% higher amounts of flavonoids expressed as quercetin equivalents. These results also correlated with the anti-radical activity of this extract, as it was higher than the activity of the ethanol extract by 25% and 52% measured in the system with DPPH and ABTS^+^ radicals, respectively (Table 1). The higher content of the compounds and the higher anti-radical activity of the F2 fraction, compared to the starting ethanol extract, indicate a substantial concentration of active ingredients per unit mass of the extract. This resulted from the preparation procedure of the extract and fractions, in which the fractionation stage was followed by drying of the sample. The subsequent series of analyses were performed on starter solutions with a standardized starting concentration of 1 mg/mL. Such a procedure gave the opportunity to compare the examined fractions and select those with the most favorable functional properties. The concentration and type of phenolic compounds depends not only on the plant species but also on its anatomical part [11,22]. Teleszko and Wojdyło [11] compared the chemical composition and antioxidant activity of extracts obtained from fruits and leaves of seven species of fruiting trees and shrubs and showed that the leaf extracts of all the tested plants were characterized by higher activity than the fruit extracts. In the case of blackcurrant, extracts from leaves showed 46% higher activity compared to fruit extracts [11]. Numerous previous studies confirm accumulation of high concentrations of flavonoids in leaves [12,23].

The accumulation in leaves is related to the physiological functions of these compounds in the plant, as they absorb radiation in the UVB range, thereby playing a protective role for the photosynthetic apparatus [5,23]. Flavonoids are compounds with indirect lipophilicity, and the best extraction yields are obtained using alcoholic solutions at concentrations of 50–80% [24]. Hence, compounds extracted with 80% ethanol in the present study eluted most efficiently to the F2 fraction. In addition, the Pearson correlation analysis showed a strong positive relationship (r^2^ = 0.945) between the sum of flavonoids and antiradical activity of the examined extracts and fractions measured in the DPPH radical system, while a significantly lower correlation was found between the sum of phenolic compounds and the antiradical activity (r^2^ = 0.479). This indicates that the flavonoids found in the F2 fraction most strongly modulate its antioxidant potential. The anti-radical activity of the blackcurrant leaf extract obtained in the present study was similar to that of ethanol extracts (70% *v*/*v*) from raspberry and blackberry leaves. This was confirmed in the study conducted by Zielonka-Brzezicka et al. [25], where the antioxidant activity of ethanol extracts from raspberry leaves was at a level of 0.71 mmol/L, i.e., 195 μg/L expressed as Trolox equivalents [25]. Therefore, it can be concluded that leaves from fruit shrubs are a rich source of active compounds with promising antioxidant properties.

### 2.2. Biological Properties

The next stage of the research consisted of the analysis of the anticancer properties of the starting extract and its fractions. The human colorectal cancer HCT 116 cell line and the prostate cancer PC-3 line were used for these analyses. The mouse fibroblast L929 line was used as a normal cell model, which is recommended for cytotoxicity studies [26,27]. The results are summarized in Figure 1. It was observed that the F2 fraction exhibited the highest cytotoxic activity towards the HCT 116 cells, and the value of IC_50_ determined for this line was 70 µg/mL, which accounted for 45% of its efficiency in the normal L929 cell line. The PC-3 prostate cancer cell line was sensitive to the extract and all its fractions. In the case of the water fraction, the observed effect was the least favorable since it equally affected the normal cells and the PC-3 cells but had a significantly weaker impact on the HCT 116 cells. The best efficiency was exhibited by the F2 fraction, for which the IC_50_ value was almost three times lower (46 µg/mL) than in the L929 line (128 µg/mL). The cytotoxic effect of the examined extracts and fractions from blackcurrant leaves correlated with their anti-radical properties (r^2^ = 0.81). The relationship between the anti-radical activity and anticancer properties of plant extracts has also been confirmed by other authors in the same PC-3 and HCT 116 cell lines exposed to sweet pepper fruit extracts [28] and soybean pod extracts [29] and in other cancer cell lines, including breast cancer (MCF7) cells treated with for soybean leaf extract [30] or colon cancer (Caco-2) cells exposed to grape peel extract [31]. Similar data were reported in scientific publications in which cytotoxicity was tested in extract fractions from medicinal and common household plants.

It is also worth mentioning that some extracts analyzed in these studies showed selective cytotoxicity towards cancer cells and no significant toxicity to normal cells (L929). However, based on the American National Cancer Institute (NCI) and Geran’s protocol, cytotoxicity can be classified as follows: IC_50_ ≤ 20 μg/mL = highly cytotoxic, IC_50_ ranging between 21 and 200 μg/mL = moderately cytotoxic, IC_50_ ranging between 201 and 500 μg/mL = weakly cytotoxic, and IC_50_ > 501 μg/mL = no cytotoxicity [32]. As shown by the results presented in Figure 1, the tested extract fractions were active in the range of 21 and 200 μg/mL against the HCT 116 and PC-3 cancer cells, i.e., they are moderately cytotoxic. Therefore, they can be considered promising sources to support classical anticancer therapy. It is estimated that 899,000 men worldwide are diagnosed with prostate cancer annually, and the number of deaths reaches 357,000 cases [33]. In turn, colorectal cancers are the fourth most common cancer in the world, making up 10% of all new cancer cases [34]. Based on these reports, the choice of prostate cancer and colorectal cancer cell lines for the present investigations seems reasonable. Previous studies indicated variable sensitivity of these cell lines to plant extracts. Using the same cell lines, Pabich et al. [29] analyzed the cytotoxic effects of fractionated soybean pod extract. They found that only the PC-3 line was sensitive to the effects of the components of the extract and its fractions, while there were no significant differences in their effect between the colorectal cancer cell line and normal cells [29]. Similarly, no cytotoxic activity in the HCT 116 line with simultaneous cytotoxic activity in the PC-3 line was exerted by extracts and fractions with variable lipophilicity obtained from the pericarp of sweet peppers [28] and hot peppers [27]. Additionally, the comparison of the anticancer effectiveness of the F2 fraction on the PC-3 line showed the highest activity of the fraction was obtained from blackcurrant leaves (64.1%), followed by hot peppers (46.7%) [27], soybean pods (41.1%) [29], and finally sweet peppers (14.4%) [28]. Therefore, it can be concluded that the F2 fraction obtained from blackcurrant leaves has the greatest potential to be used as a component of the diet in the prevention of the development of these diseases.

### 2.3. Chromatographic Analyses

LC-QTOF-MS analyses were performed to estimate the qualitative composition of the tested extract and its fraction. The presence of twelve compounds was revealed by the analysis of mass spectra of their standards. For these compounds, the LC-MS analyses were performed in the *scan* mode to confirm their retention times and in the *target* mode to confirm their fragmentation spectra. Other compounds were identified on the basis of literature data and available databases (PubChem and ChemSpider) [22,35,36]. Finally, 30 polyphenolic compounds were identified in the examined samples and listed in Table 2 with numbering according to the retention time. Taking into account the chemical structure, the compounds were identified and divided into three groups: phenolic acids, flavonoids, and other phenolic compounds. Eight phenolic acids (**3**, **6**, **7**, **8**, **9**, **11**, **18**, and **20**) and 19 flavonoids, including aglycones of myricetin (**10**), naringenin (**26**), and quercetin (**30**), were identified. The other flavonoids were *O*-glycosides of quercetin (**13**, **15**, **16**, **19**, **23**, and **27**), kaempferol (**14**, **17**, **22**, **24**, **28**, and **29**), isorhamnetin (**2**, **21**, and **25**), and myricetin (**12**). The last three compounds were classified as other phenolic compounds: *p*-coumaric acid glucose ester (**4**) and two compounds from the catechin group (**1** and **5**) (Table 2). Most of the identified compounds have previously been detected in blackcurrant leaf extracts [35,36,37,38]. In the phenolic compound profile obtained in the present work, 17 compounds were identical to those reported by D’Urso et al. [36], who identified 31 phenolic compounds in 3 types of blackcurrant leaf extracts. The main differences lay in the presence of anthocyanins, which were not identified in the present study (Table 2). Similarly, in comparison with the results published by Piotrowski et al. [37], who identified 39 compounds in methanol leaf extracts from 11 blackcurrant cultivars, the qualitative analysis performed in the present study detected 18 compounds. The chemical composition of blackcurrant leaves is determined by many factors, including the variety [37], habitat conditions (climate, soil type) [17], and harvest date [13,39]. The method of extraction and the research methods used are also important. Using three types of solvents (water, methanol, and ethanol–water 1:1) D’Urso et al. [36] found that ethanol extracts exhibited the greatest accumulation of flavonoids. In the present work, a solution of ethanol in water (80% *v*/*v*) was the starting solvent used for the preparation of extracts, which were separated into three fractions after evaporation. The LC-MS analysis of the fractions indicated that some compounds were present in only one fraction and some were detected in two or three fractions (Table 2). Typically, compounds with lower retention times were present in the F1 fraction and those with higher retention times were contained in the F3 fraction. This demonstrates the efficiency of solid-phase extraction in the separation of multicomponent matrices. In addition, not all compounds eluted in accordance with the lipophilic character. To illustrate these differences, Table 2 includes a column with values of the partition coefficient (logP) taken from the PubChem database. The partition coefficient, i.e., the ratio of compound concentrations between the organic and aqueous phases (*n*-octanol–water), is often used in QSAR studies, which allow computational prediction of the biological properties of organic compounds [40].

Over the past two decades, a number of algorithms have been used to make this kind of calculation. The XlogP3 algorithm was used in this database [41]. According to the logarithm definition, the logP = 1 value indicates that the concentration of the compound in the organic phase is 10 times higher than in the aqueous phase, and an increase in the logP value is associated with stronger lipophilic properties of the compound [40,41]. The values of the partition coefficients of compounds identified in the ethanol extract and its subfractions ranged from −2.8 to 2.4 for compounds **2** as well as **11** and **26**, respectively (Table 2). Compounds **11** and **26**, which had the same logP value, had significantly different retention times in the chromatographic analysis. It can therefore be concluded that the partition coefficient is not a determinant of the separation of compounds in the chromatographic process. It seems that the spatial properties and concentration of the compound in the starting extract had a more significant impact on the separation. In turn, the analysis of the chemical composition of the obtained fractions revealed the absence of compounds with logP values > 1 in the F1 fraction, i.e., they did not elute with water. However, in the case of fractions F2 and F3, no clear relationship was found between their presence in these fractions and the values of the partition coefficient for two reasons. Firstly, fractionation by solid phase extraction is not a specific technique; hence, the extracts were characterized by a variable composition and different anti-radical and anticancer activity (Table 1 and Table 2). Secondly, the LC-QTOF-MS analysis is a qualitative analysis and, as mentioned above, the concentration of the compounds significantly affects the elution processes in solid phase extraction and in chromatography.

Therefore, the HPLC-DAD analysis was carried out to determine the quantitative differences between the obtained fractions. The study was conducted on 12 compounds identified in the ethanol extract and having available standards. The chromatogram of the ethanol extract indicating compounds analyzed quantitatively is shown in Figure 2.

The content of the compounds in the individual fractions is expressed in mg/g extract based on the equations of calibration curves prepared for each standard (Appendix A). The results of the quantitative analysis are summarized in Table 3. The ethanol extract was dominated by two acids (**6** and **9**) and two quercetin *O*-glycosides (**15** and **19**). Their total content constituted 63% of the mass of the quantified compounds. A qualitative–quantitative HPLC-DAD analysis of phenolic compounds present, among others, in blackcurrant leaves was carried out by Tian et al. [42]. The qualitative profile of phenolic compounds detected by these authors contained flavonol glycosides, mainly quercetin and kaempferol derivatives, which accounted for 70–90% of all phenolic compounds. The highest concentrations were exhibited by quercetin 3-*O*-(6″-malonyl) -glucoside, which also dominated in red currant leaves, and the content of this compound in blackcurrant leaves was 169 mg/100 g [42]. In the present study, this compound was the dominant component of the ethanol extract as well, and its concentration in 1 g of dry extract was 2.085 ± 0.006 mg (Table 3). The quantitative analysis of 12 phenolic compounds in the three fractions coincided with their qualitative analysis. The F1 fraction, in which four components were determined with a total mass of 51.706 mg in 1 g of extract, had the lowest content of the compounds determined quantitatively. The low content of phenolic compounds in the F1 fraction was also confirmed by previous spectrophotometric analyses (Table 1). This fraction contained the lowest amounts of TP and TF and showed practically no anti-radical activity. In contrast, the highest level of phenolic compounds was detected in the F2 fraction, in which nine components constituting a total of 139.913 mg in 1 g of dry extract were determined quantitatively. This fraction was also characterized by the best parameters in the spectrophotometric analyses (Table 1) and showed the strongest anticancer properties against both cell lines (Figure 1). The dominant compound of this fraction was quercetin-3-*O*-glucoside, whose mass accounted for almost 38% of the mass of the quantified components. In the F3 fraction, eight compounds were quantitatively determined, with quercetin-3-*O*-malonylglucoside as the dominant compound constituting almost 81% of the compounds quantified in this fraction (Table 3). The division of the starting ethanol extract into three subfractions yielded fractions containing a condensed amount of active compounds in a relatively small mass. This was particularly evident in the F2 fraction, where flavonoids (Table 3) constituted a substantial part. These compounds with indirect lipophilicity eluted with high efficiency in the 40% solution of methanol in water. The fractionation of ethanol extracts seems to be a good method to obtain enriched plant extracts with promising antioxidant and anticancer potential. Nevertheless, it should be mentioned that, in addition to the quantitatively determined compounds, the extract and its fractions contained other compounds that could not be quantitatively determined in this study but may have affected the properties of the analyzed samples.

The research results presented here as well as our previous study on the fractionation of sweet and hot pepper extracts [27,28] and soybean pod extracts [29] indicated a definite potential of the fraction eluted with the 40% solution of methanol in water, as it contained the highest amounts of mixtures of flavonoids and phenolic acid derivatives, which can be successfully used as natural food additives instead of synthetic antioxidants. Their anticancer properties open the way to the design of both functional foods and dietary supplements that can be used in the prevention of cancer development.

The anticancer activity of phenolic compounds results from several mechanisms of action: (i) induction of cell apoptosis, (ii) an antioxidant effect on cell components, and (iii) an anti-inflammatory effect. (i) Apoptosis is cell death programmed precisely at the genetic level. Its deregulation is considered one of the characteristics of cancer progression, which causes mutant cells to multiply [43]. Therefore, the induction of apoptosis is crucial in the removal of precancerous lesions. A number of studies indicate that both pure phenolic compounds and natural plant extracts rich in these compounds can restore the sensitivity of cancer cells to apoptosis [43,44]. For example, it has been shown that bergamot and its extracts have anticancer effects [44]. These effects were explained by the presence of flavonoids, which had the ability to disrupt the main stages of cancer development, i.e., cancer induction, promotion, and progression. A very extensive study [45] conducted in vitro on human cancer cells of the nervous system (SH-SY5Y, PC12), prostate (PC3), and breast (MDAMB-231) showed that treatment with bergamot juice at various concentrations (1–5%) stopped the progression of the disease. Moreover, the juice showed the ability to reduce the growth rate of various cancer cell lines, the mechanisms of which depended on the type of cancer [46,47]. In another study on human colon cancer cells, it was shown that low concentrations of bergamot juice induced inhibition of mitogen-activated protein kinase (MAPK)-dependent pathways and elicited cell cycle arrest and alteration of apoptosis, while its high concentrations caused oxidative stress causing DNA damage [48]. (ii) Natural cellular metabolism includes oxidation and reduction reactions producing reactive oxygen/nitrogen species. They are involved in some regulatory processes, including gene expression, cell proliferation, and apoptosis. When reactive species are generated in quantities that exceed the antioxidant capacity of cells, they may pose a threat to these cells. They can oxidize biological macromolecules such as DNA, proteins, lipids, carbohydrates, and enzymes, which may ultimately change their functions [44]. For example, high concentrations of reactive oxygen species interact extensively with nuclear DNA, causing mutations and genome instability [49,50]. Through reaction with proteins, they form protein adducts, and their reaction with lipids contributes to damage to cell membranes. Oxidative stress damage has been found to occur in disease states and cancer cells. The body contains endogenous antioxidants that are supposed to eliminate excess concentrations of free radicals in cells. However, these compounds often do not provide full protection. For this reason, exogenous antioxidants supplied from food, supplements, and pharmaceuticals are important. In recent years, research has focused particularly on the ability of natural products to reduce cellular oxidative stress [51]. The antioxidant activity of natural plant products is a result of the interaction of their ingredients, which may have a synergistic or antagonistic effect on each other. However, due to the complex composition of natural extracts, their antioxidant potential can only be determined experimentally. Unlike in the case of mixtures, the mechanism of the antioxidant activity of many pure compounds has been demonstrated at the cellular and structural levels [50]. The antioxidant activity of phenolic compounds is related to the presence of at least one catechol (1,2-dihydroxybenzene) moiety in their structure, which can be oxidized in a reaction with free radicals, eliminating the formation of further free radicals and preventing lipid peroxidation [50,51]. At the cellular level, the antioxidant activity of phenolic compounds is associated with the ability to induce the activity of enzymes that detoxify reactive oxygen species, including superoxide dismutase (SOD), catalase (CAT), glutathione S-reductase (GSR), and glutathione S-transferase (GST) [44]. (iii) The links between inflammation and cancer development were confirmed after the discovery of leukocytes in neoplastic tissues at the end of the 19th century. Currently, the relationship between the occurrence of inflammation and the development of cancer is explained by two paths. The first is internal, where genetic events determine the development of cancer, which influences the formation of an inflammatory microenvironment. The second one is the external pathway. It begins with inflammation and becomes chronic, which ultimately leads to the development of cancer [52]. Many studies analyzing the effect of polyphenol-rich plant extracts on slowing down inflammation induced in experimental animals (mice or rats) confirmed their beneficial effect on slowing down this process through multidimensional action. At the cellular level, they can reduce mRNA levels in IL-8 cells treated with TNF-α [53]. There was also a significant reduction in skin inflammation associated with the administration of a therapeutically natural plant extract, which was associated with a reduction in intercellular adhesion molecule 1 (ICAM-1) with inducible nitric oxide synthase (iNOS) and nitric oxide (NO) [54].

Literature data indicate that single phenolic compounds often exhibit multidirectional biological activity, e.g., quercetin-3-*O*-rutinoside with its antioxidant and anti-inflammatory properties and thus anticancer activity [55]. For example, in vitro studies have demonstrated its antitumor activity against human colorectal cancer cell lines (HCT 116) [56] and PC-3 cancer cells [57]. Similarly, chlorogenic acid is a widespread compound with proven pharmacological properties. In vitro and in vivo studies have shown that its main pharmacological action is based on antioxidant, antibacterial, and antiviral properties [58]. In turn, in a study conducted by Luo et al. [59], rosmarinic acid was shown to inhibit the proliferation of oral cancer cells, and the level of inhibition was dose-dependent. The antiproliferative role of rosmarinic acid consisted in induction of apoptosis and cell cycle arrest in the G2/M phase in oral cancer cells. Treatment with rosmarinic acid also caused endoplasmic reticulum stress and had a negative effect on the migration potential of cancer cells in a concentration-dependent manner [60]. *p*-Coumaric acid isolated from *B. dracunculifolia* had healing effects against acetic acid-induced gastric ulcers in rats, stimulated the proliferation of mouse fibroblast cells, and reduced the viability of human gastric adenocarcinoma (AGS) cells [61]. As reported by Gao et al. [62], ferulic acid can significantly inhibit the proliferation and invasion of cervical cancer cells. Studies on the cytotoxicity of this acid against human osteosarcoma, glioma (U87MG), and prostate cancer cell lines showed that ferulic acid induced cytotoxicity against these cells [62]. Flavonoid *O*-glycosides in the free state also showed varied antitumor activity. For example, quercetin-3-*O*-glucoside inhibited the growth of pancreatic cancer cells [63] but exhibited cytotoxicity to HCT 116 cancer cells [64]. Studies on the effect of luteolin-7-*O*-glucoside on the migration and invasion of oral cancer cells showed that, in addition to preventing their proliferation, this compound caused a significant reduction in the migration and invasion of these cancer cells. Furthermore, it reduced cancer metastasis by reducing p38 phosphorylation and lowering matrix metalloproteinase (MMP)-2 expression [65]. To summarize, phenolic compounds contained in natural plant extracts and their individual components demonstrate multidirectional biological activity, which can ultimately be observed as an anticancer effect that is beneficial to health. Moreover, many experimental studies have proven that these compounds have a unique ability to distinguish cancer cells from normal cells, inhibiting proliferation and inducing apoptosis only in cancer cells [44]. Additionally, recent research indicates the possibility of synergistic effects between phenolic compounds, which may result in higher biological activity of natural mixtures compared to the activity of individual compounds [66]. However, the information on this subject is still incomplete, probably due to the difficulties in determining the exact chemical composition of natural plant extracts and obtaining extracts with a repeatable composition.

Another issue that should be taken into account when considering phenolic compounds as anticancer agents is their bioavailability. The bioavailability of any compound taken with food is influenced by its digestion, absorption, and metabolism. For this reason, there is no correlation between the amount of phenolic compounds administered with food and their bioavailability in the human body. Polyphenols undergo enzymatic cleavage after consumption. The carbohydrate part (if present) and its aglycones enter the small intestinal epithelial cells by passive diffusion [67]. If polyphenolic compounds have not been absorbed in this part of the digestive tract, they enter the large intestine where they are metabolized by the microflora. They can therefore partially change the intestinal microflora, acting as prebiotics [68]. After absorption, the final derivatives undergo methylation, sulfation, and glucuronidation reactions and reach the liver with the blood, where they can be subjected to phase II metabolism, transported to appropriate tissues, and finally excreted [67]. Based on this information, it can be concluded that the beneficial effect of polyphenols on health depends on both the amount consumed and their bioavailability [69]. The bioavailability of polyphenols varies among different classes and depends on the chemical structure. Based on a number of studies focused on this issue, the following bioavailability scale was constructed: phenolic acids > isoflavones > flavonols > catechins > flavanones and proanthocyanidins > anthocyanins [44]. In the present study, the results of the quantitative analysis of the F2 fraction indicate the presence of four phenolic acids, and the qualitative analysis confirmed the presence of two more compounds of this group. Therefore, given the information cited above, the F2 fraction seems to be a promising remedy helpful in the treatment of cancer in terms of both its biological activity and the bioavailability of its components. The available literature studies indicate that this type of extract has not been tested so far in terms of bioavailability. Taking into account their quite complex chemical composition, it is not possible to clearly determine which compounds will be absorbed in the upper gastrointestinal tract and which will reach the large intestine, affecting its microflora. Additionally, especially in the case of prostate cancer (PC-3), additional in vivo analyses are necessary to determine the type and concentration of F2 fraction metabolites in the blood after absorption.

### 2.4. PCA Analysis

An important element of the comprehensive assessment of parameters describing antioxidant and anticancer properties as well as the chemical composition of the analyzed extracts and fractions is their mutual correlation. In order to clarify the structure of these variations, a principal component analysis (PCA) was carried out, and the calculation of the components was made on the basis of the correlation matrix. To investigate the relationship between the chemical composition of the extract and its fractions and their antioxidant and anticancer properties, a scaled PCA analysis was applied to the entire data matrix consisting of rows expressing the type of extract and columns containing the values of individual determinations. Nineteen variables were used, DPPH, ABTS, L929, HCT 116, and PC-3, as well as the determined amounts of phenolic compounds: gallic acid, chlorogenic acid, neochlorogenic acid, ferulic acid, *p*-coumaric acid, rosmarinic acid, quercetin-3-*O*-rutinoside, quercetin-3-*O*-glucoside, quercetin-3-*O*-galactoside, kaempferol-7-*O*-glucoside, quercetin-3-*O*-malonylglucoside, and quercetin. The reduction in the dimensions to two was performed based on the scree plot. The PCA analysis (Figure 3A) explains 88.1% of the variability of the first two main components: 63.57% and 24.44%, respectively. The statistical analysis of the results showed that the anticancer activity of the extracts and fractions against the HCT 116 cancer cells was largely determined by the presence of quercetin derivatives (compounds **13**, **15**, and **16**), two phenolic acids (**11** and **20**), and kaempferol-7-*O*-glucoside (**22**). In order to confirm these results, Spearman’s correlation coefficients were additionally determined (Appendix A), which confirmed high correlations between the above-described data. The correlation coefficients take negative values because one of the components of the analysis (anticancer activity) is expressed as IC50. The tests of biological activity against the prostate cancer cells (PC-3) showed anticancer properties of all analyzed samples against this cell line, but there was no significant impact of the quantitatively determined compounds on the above-mentioned biological activity; probably, other compounds, which were not quantified in this study, were responsible for the analyzed properties of the extract and fractions in the black currant leaves. Figure 3B shows the projection of cases (extract and fractions) on the plane of the main components, which illustrates the similarity between the studied fractions. The mutual arrangement of the analyzed cases in relation to each other indicates that the F1 and F3 fractions show similar activities towards the analyzed radicals and the tested cell lines. The F2 fraction was characterized by different properties that were the subject of consideration. Comparing the results presented in Figure 3A,B, it can be seen that the location of the F2 fraction cases correlates with the high content of phenolic compounds in this fraction.

## 3. Materials and Methods

### 3.1. Plant Material and Extraction

Blackcurrant (*Ribes nigrum* L.) leaves were collected from private cultivation in the Lublin region. The leaves were washed, dried, and cut into smaller pieces. The material was frozen (−18 °C) and freeze-dried. Sublimation drying was carried out for 72 h using a freeze dryer (Free Zone 12, Labconco Corporation, Kansas City, MO, USA) at −80 °C and 0.04 mbar.

The lyophilized material was crushed and extracted with 80% ethanol (30 min) using ultrasound-assisted extraction (10 min). Eighty percent ethanol was chosen for the extraction procedure because it is the most effective solvent for the elution of phenolic compounds [24]. Then, the extract was filtered and concentrated to a volume of 1 mL using a vacuum evaporator (Büchi, Flawil, Switzerland) at 40 °C. The concentrated extract was divided into three fractions with variable hydrophilicity by solid phase extraction (SPE): aqueous, 40% methanol/water, and 70% methanol/water. The division was carried out on SPE-C18 columns (5 g, Sep-Pak, Waters, Milford, MA, USA). The ethanol extract and fractions were concentrated and finally lyophilized. Solutions with concentrations of 1 mg/mL were prepared from each lyophilizate prior to the analysis. Samples dedicated for microbiological analyses were dissolved in DMSO, and samples for phytochemical analyses were dissolved in 80% ethanol, water, and 40% and 70% methanol for the ethanol extract, the water fraction, and the methanol fractions, respectively.

### 3.2. Chemical Composition and Biological Activity

#### 3.2.1. Total Phenolic Compounds (TP)

The total content of phenolic compounds was determined with the Folin–Ciocalteu method [70]. The reaction mixture consisted of 60 μL of the tested extract, 1.5 mL of Folin reagent diluted in water at a ratio of 1:10, 1.2 mL of sodium bicarbonate (75 g/L), and 0.54 mL of distilled water. The reaction mixture was stored at room temperature for 30 min. The absorbance was measured at a wavelength λ = 760 nm. The results were expressed as chlorogenic acid equivalent μg/mL.

#### 3.2.2. Total Flavonoids (TF)

The sum of flavonoids contained in the extracts was determined using the AlCl_3_ method [71]. The reaction mixture consisted of 0.5 mL of the tested extract, 1.5 mL of 96% ethanol, 0.1 mL of AlCl_3_ (10%), 0.1 mL of sodium acetate (1 M), and 2.8 mL of distilled water. The reaction mixture was stored at room temperature for 40 min. The absorbance was measured at a wavelength of λ = 415 nm. The final results were calculated on the basis of a calibration curve prepared for quercetin and expressed as μg quercetin/mL.

#### 3.2.3. Antiradical Activity (AA)

Antiradical activity was determined with two methods: in the system with the 1,1-diphenyl-2-picryl hydrazyl radical (DPPH^•^) dedicated to lipophilic compounds and in the system with the hydrophilic radical-cation ABTS^+•^ (2,2′-azino-bis (3-ethylbenzothiazoline-6-sulfonic acid). The analyses in the DPPH^•^ system were carried out with the method described by Gođevac et al. [72] The reaction mixture consisted of 100 μL of the extract and 4 mL of a 0.1 mM methanolic solution of DPPH^•^. The reaction mixture was stored at room temperature for 30 min, and then the absorbance was measured at a wavelength of λ = 515 nm. The results were expressed as the Trolox equivalent (μg Trolox/mL). The determinations in the ABTS^+^ radical system were made using the methodology described by Re et al. [73]. The ABTS^+•^ cation was synthesized in a reaction of an aqueous solution of ABTS (7 mM) with potassium persulfate (2.45 mM), which was stored in the dark for 12 to 16 h. The starting solution was diluted to an absorbance of 0.7 (±0.02) at λ = 734 nm. The reaction mixture contained 3 mL of the ABTS^+•^ solution and 20 μL of the extract. In the control sample, the sample was replaced by ethanol. The reaction mixture was stored at room temperature for 10 min and then measured at λ = 515 nm. The results were presented as the equivalent of the Trolox antioxidant potential in the reaction with the cation radical on the basis of a calibration curve prepared for this compound and expressed in µg of Trolox/mL.

#### 3.2.4. Anticancer Properties

The cytotoxicity of the extract and the variable lipophilicity fraction was assessed using two tumor cell lines: human prostate cancer (PC-3) and colon cancer (HCT 116). The L929 mouse fibroblast line was used as a reference for normal cells. The in vitro cytotoxicity studies were carried out in accordance with EN ISO 10993-5:2009 [74]. The HCT 116, PC-3, and L929 cell lines were obtained from the American Type Culture Collection (ATCC, Rockville, MD, USA). The PC-3 cell line was cultured in DMEM-F12 medium, while the HCT 116 cells were kept in RPMI-1640 medium. The reference cells were grown in IMDM medium. All media used were supplemented with fetal bovine serum (10%) and 1% penicillin/streptomycin. The cells were grown in an incubator at a constant temperature of 37 °C in an atmosphere saturated with water vapor with a 5% addition of CO_2_. Regular two-week passages were performed with 0.025% trypsin/EDTA after the cells reached 90% of full coverage of the vessel surface with a single layer of cells (confluence). The PC-3 and HCT 116 cell lines were tested for mycoplasma contamination. MTT is a quantitative colorimetric method used to determine the state of cellular metabolism after treatment with tested compounds. It is used to assess the cytotoxic effects of chemicals on different cell types [75]. MTT [3-(4,5-dimethyl-2-thiazolyl)-2,5-diphenyl-2H-tetrazolium bromide] is converted into insoluble formalate when dissolved. This product is impermeable to cell membranes and accumulates in metabolically active cells. The test was optimized for the cell lines and chemicals used in these experiments. The cancer cells (PC-3, HCT 116) and mouse fibroblasts (L929) were cultured for 24 h (37 °C, 5% CO_2_) in 96-well microplates. The cells were then incubated in the same conditions with freshly prepared extracts for another 72 h.

The MTT solution (5 mg/mL) was added at the end of incubation. The contents of the wells were dissolved by the addition of 100 μL DMSO. The absorbance was measured spectrophotometrically using a microplate reader (BioTek Instruments, Inc., Winooski, VT, USA) at λ = 570 nm, with DMSO used as a blank. The results were analyzed in GraphPad Prism version 7.0 (GraphPad Prism Software Inc., San Diego, CA, USA) and presented as IC_50_ values. Dose–response curves were generated by graphing the cell growth percentage against the logarithmic concentration of the compound. The IC50 (concentration that inhibits 50%) values were determined through a non-linear regression model applied to the sigmoidal dose–response curve. The results were presented as the mean ± SEM repeats.

### 3.3. Chromatographic Analyses

#### 3.3.1. LC-MS Analysis

The qualitative analysis of the extract and its fraction was carried out using the LC-QTOF-MS method on an Agilent Technologies 1290 series liquid chromatograph coupled with a high-resolution Agilent Technologies 6530 Q-TOF LC/MS mass spectrometer (Agilent Technologies, Palo Alto, CA, USA). The chromatographic separation was carried out on column C18 (Agilent, Zorbax Extend C18, dimensions: 2.1 × 50 mm and grain size: 1.8 mm). The mobile phase consisted of 0.1% acetic acid in acetonitrile (A) and in water (B). The separation analysis was performed in a gradient pattern with the following concentration of solvent A: 0–5 min: 20–45%, 5–10 min: 45%, and 10–15 min: 95%; after 15–16 min of the program, the composition of the eluent returned to the initial gradient. The column was balanced for 3 min before the next injection. The mobile phase flow was 0.4 mL/min, and the volume of the dispensed sample was 5 μL. Mass spectra were obtained in the mass range of 100–2000 Da with a scan time of 1.0 s in the positive ionization mode (ESI+). The following parameters were used: capillary voltage 3500 V, nitrogen temperature 300 °C at 5 L/min flow rate, shielding gas temperature 300 °C at 8 L/min flow, and nebulizer pressure at 35 psi. Data were collected using “MassHunter Acquisition” and “MassHunter Qualitative Analysis” software 10.0 (Agilent Technologies, Inc., Santa Clara, CA, USA). The “Personal Compound Database (PCD) and Library Software B.08.00” system was used for the direct search in the database and library, where the compound was identified using the Find Compound by Formula (FBF) algorithm. The identification of compounds in this algorithm is based on the analysis of isotopic patterns. When an isotopic standard is matched, the deviation (ppm) between the isotopes and the monoisotopic mass is taken into account. The contribution to the overall score was determined as follows: mass score 100, isotope abundance score 60, isotope spacing score 50, and retention time 100 [28].

#### 3.3.2. HPLC-DAD Analysis

The HPLC method was used for the quantitative determination of 12 main compounds present in the extract and fractions from the blackcurrant leaves. These included gallic acid, chlorogenic acid, neochlorogenic acid, *p*-coumaric acid, quercetin-3-*O*-rutinoside, ferulic acid, kaempferol-7-*O*-glucoside, quercetin-3-*O*-glucoside, quercetin-3-*O*-galactoside, quercetin-3-*O*-malonylglucoside, rosmarinic acid, and quercetin (Sigma-Aldrich, Steinheim, Germany. The analysis was performed using an Empower-Pro chromatograph (Waters), which consists of a quaternary pump (M2998 Waters) with a degasser and a UV-Vis diode array detection (DAD) system. The separation was performed on a column filled with modified silica gel RP-18 (Atlantis T3—Waters, 3 μm, 4.6 mm × 150 mm). The mobile phase consisted of solvent A (acetonitrile) and B (1% acetic acid in water). The identification of the investigated compounds was performed on the basis of retention times and diode array spectral characteristics in comparison with available standards. Quantitative analyses were performed based on the areas of the peaks of the tested compounds and calibration curves prepared separately for each standard compound (Appendix A). For quantitative analysis, the DAD detection was conducted at 280 nm for phenolic acid derivatives and at 330 nm for flavonoid derivatives [28].

### 3.4. Statistical Analyses

The results were obtained in three repetitions, and the data were expressed as their average ± SD. The significance of differences between the means was determined by a one-way ANOVA LSD test with a 5% probability of error. To investigate the relationship between the chemical composition of the extract and its fractions and their bioactivity, a principal components analysis (PCA) was applied. Additionally, to confirm the correlations between the obtained results, Spearman’s correlation coefficients were determined. All calculations were performed using the Statistica 13.1 PL; StatSoft Inc., Tulsa, OK, USA.

## 4. Conclusions

The present study confirmed that blackcurrant leaves are a rich source of phenolic compounds with high antioxidant activity and anticancer properties. Higher bioactivity of the obtained fractions compared to the activity of the starting extract was demonstrated as well. This proves the validity of the hypothesis proposed previously. In addition, the separated F2 fraction deserves special attention due to its high anticancer potential against PC-3 and HCT 116 cancer lines and strong antioxidant properties, which gives the possibility of its versatile use. However, the tested fractions were found to exert different biological activities against the two cancer lines. In the case of the HCT 116 line, the statistical analyses indicate the dominant effect of compounds **11**, **13**, **15**, **16**, **20**, and **22** on anticancer activity. On the other hand, these relationships were not so clear in the case of the PC-3 line. Based on the present results, it can be concluded that the blackcurrant leaf extract and its fractions are a promising source of active compounds that can be used in the pharmaceutical industry and as natural food additives. They can also be an attractive alternative to synthetic antioxidants and dietary supplements. However, further research is needed to determine their bioavailability and identify markers for the quality control of blackcurrant extracts before use.

## Figures and Tables

**Figure 1 molecules-28-07459-f001:**
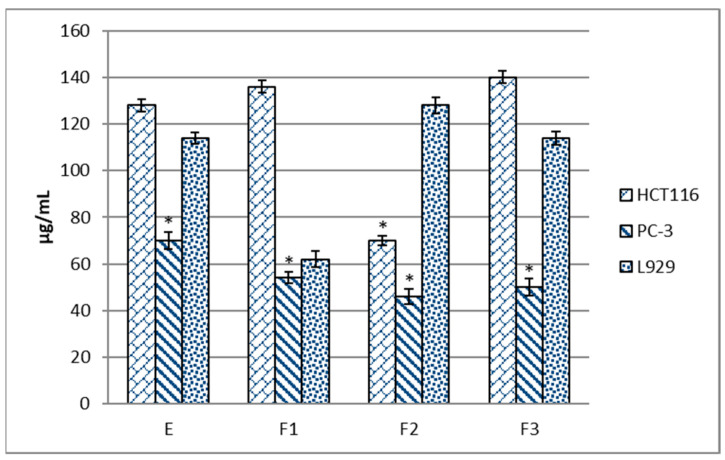
Anticancer properties of the ethanol extract from blackcurrant leaves and its subfractions expressed as IC_50_ (μg/mL). The values are expressed as the mean ± SD (*n* = 3). According to the one-way ANOVA LSD test, means with a *p*-value lower than 0.05 were considered statistically different. Values marked by * mean cytotoxic activity. E—ethanol extract; F1—water fraction, F2—40% methanol/water fraction; and F3—70% methanol/water fraction.

**Figure 2 molecules-28-07459-f002:**
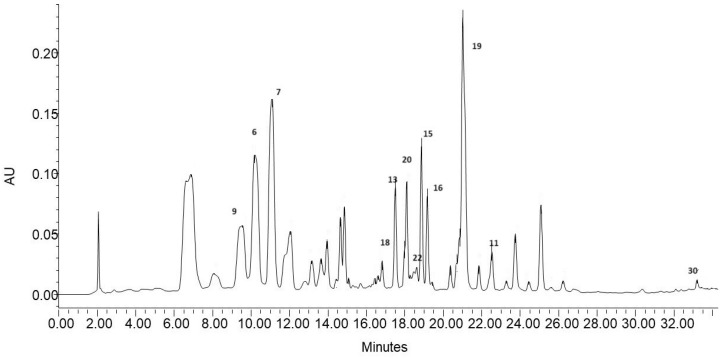
Chromatogram of the blackcurrant leaf ethanol extract (λ = 330 nm). The peaks are numbered as described in Table 2 and Table 3.

**Figure 3 molecules-28-07459-f003:**
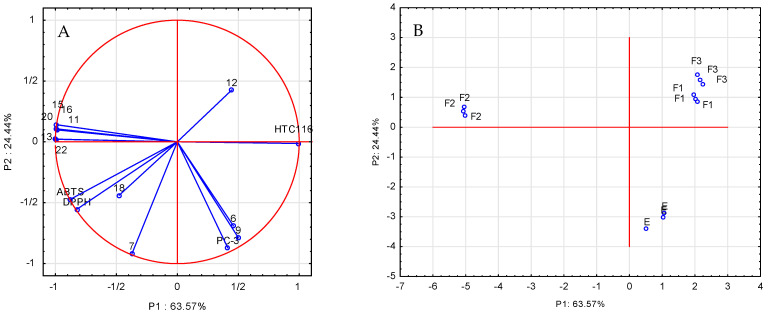
Analysis of the main components (PCA) of selected phenolic compounds examined in the blackcurrant leaf extract: (**A**) projection of variables on the plane of factors; (**B**) projection of cases: E—ethanol extract, F1—water fraction, F2—40% methanol/water fraction, and F3—70% methanol/water fraction.

**Table 1 molecules-28-07459-t001:** Total phenolic compounds (TP), total flavonoids (TF), and antiradical activity (AA) of the ethanol extract and its fraction determined in the system with DPPH and ABTS^+^ radicals.

	TP ^1^	TF ^2^	AA_DPPH_ ^3^	AA_ABTS_ ^3^
E	2055.62 ^b^ ± 52.07	0.3355 ^b^ ± 0.005	230.73 ^b^ ± 0.47	368.72 ^b^ ± 8.7
F1	864.33 ^c^ ± 26.03	0.0769 ^d^ ± 0.004	43.81 ^c^ ± 1.71	ppd ^4^
F2	2532.88 ^a^ ± 144.96	0.6282 ^a^ ± 0.004	288.07 ^a^ ± 1.03	561.84 ^a^ ± 3.49
F3	905.67 ^c^ ± 45.56	0.1378 ^c^ ± 0.005	45.32 ^c^ ± 2.17	ppd ^4^

The values are expressed as the mean ± SD (*n* = 3). According to the one-way ANOVA LSD test, means with a *p*-value lower than 0.05 were considered statistically different. Different letters in the same column indicate a significant difference between the results. E—ethanol extract; F1—water fraction; F2—40% methanol/water fraction; F3—70% methanol/water fraction; and ^1^ TP: chlorogenic acid equivalent μg/mL, ^2^ TF: quercetin equivalent μg/mL, ^3^ AA: Trolox equivalent μg/mL, and ^4^ ppd—result below detection level.

**Table 2 molecules-28-07459-t002:** Profile of phenolic compounds in the blackcurrant leaf ethanol extract and its fractions determined with the LCMS-ESI-QTOF method.

No	R_t_	Name *	Molecular Formula	M_obs_	M_teor_	*m*/*z*	E	F1	F2	F3	XlogP3 **	Ref.
**1**	0.522	Epigallocatechin	C_15_H_14_O_7_	306.074	306.0739	307.0813	+	+	+	+	0	[40]
**2**	0.539	Isorhamnetin-3-*O*-Rut-7-*O*-Glc	C_34_H_42_O_21_	786.2206	786.2219	787.2307	+	+	-	+	−2.8	[41]
**3**	0.559	3-Hydroxybenzoic acid	C_7_H_6_O_3_	138.0313	138.0317	139.0386	+	-	+	-	1.5	[42]
**4**	0.575	*p*-CoumaroylGlc	C_15_H_18_O_8_	326.1002	326.1002	327.1136	+	+	+	+	−0.8	[41]
**5**	0.575	Epicatechin	C_15_H_14_O_6_	290.0807	290.0790	291.0881	+	-	+	+	0.4	[41]
**6**	0.58	Chlorogenic acid	C_16_H_18_O_9_	354.0956	354.0951	377.0833	+	+	-	+	−0.4	standard
**7**	0.58	Neochlorogenic acid	C_16_H_18_O_9_	354.0937	354.0951	355.1022	+	+	+	+	−0.4	standard
**8**	0.58	*p*-Coumaroylquinic acid	C_16_H_18_O_8_	338.1002	338.1002	339.312	+	+	+	+	−0.1	[40]
**9**	0.642	Gallic acid	C_7_H_6_O_5_	170.0221	170.0215	171.0293	+	+	-	+	0.7	standard
**10**	0.656	Myricetin	C_15_H_10_O_8_	318.0383	318.0376	319.0453	+	-	-	+	1.2	[42]
**11**	0.659	Rosmarinic acid	C_18_H_16_O_8_	360.0845	360.0845	361.0912	+	-	+	-	2.4	standard
**12**	0.696	Myricetin 3-*O*-Gal	C_21_H_20_O_13_	480.0912	480.0904	481.0983	+	-	+	+	0	[26]
**13**	0.739	Quercetin-3-*O*-Rut	C_27_H_30_O_16_	610.1543	610.1534	633.1428	+	-	+	-	−1.3	standard
**14**	0.796	Kaempferol-di-hexoside	C_27_H_30_O_16_	610.1582	610.1534	611.1649	+	-	-	+	−1.1	[40]
**15**	0.842	Quercetin-3-*O*-Glc	C_21_H_20_O_12_	464.0964	464.0955	303.0497	+	-	+	+	0.4	standard
**16**	0.913	Quercetin-3-*O*-Gal	C_21_H_20_O_12_	464.0957	464.0955	465.1031	+	-	+	+	0.4	standard
**17**	0.939	Kaempferol-3-*O*-Rut	C_27_H_30_O_15_	594.1589	594.1585	595.1654	+	+	-	+	−0.9	[25,26]
**18**	0.946	p-Coumaric acid	C_9_H_8_O_3_	164.0475	164.0473	165.0547	+	-	+	+	1.5	standard
**19**	0.992	Quercetin-3-*O*-malonyl-Glc	C_24_H_22_O_15_	550.0963	550.0959	551.1036	+	+	+	+	0.1	standard
**20**	1.022	Ferulic acid	C_10_H_10_O_4_	194.0578	194.0579	195.065	+	-	+	-	1.5	standard
**21**	1.096	Isorhamnetin-3-*O*- Rut	C_28_H_32_O_16_	624.1688	624.169	625.1792	+	-	-	+	−1	[43]
**22**	1.112	Kaempferol -7-*O*- Glc	C_21_H_20_O_11_	448.101	448.1006	471.09	+	+	-	+	0.7	standard
**23**	1.112	Quercetin-3-*O*-Ara	C_20_H_18_O_11_	434.0878	434.0849	435.0354	+	+	-	+	0.4	[25]
**24**	1.246	Kaempferol-3-*O*-Gal	C_21_H_20_O_11_	448.1012	448.1006	449.1082	+	+	-	+	0.7	[26]
**25**	1.246	Isorhamnetin-3-*O*-Glc	C_22_H_22_O_12_	478.1133	478.1111	479.1197	+	-	-	+	0.7	[40]
**26**	1.388	Naringenin	C_15_H_12_O_5_	272.069	272.0685	273.0762	+	-	-	+	2.4	[41]
**27**	1.412	Quercetin-3-*O*-acetylGlc	C_23_H_22_O_13_	506.106	506.106	507.1154	+	-	-	+	0.4	[40]
**28**	1.495	Kaempferol-3-*O*-malonyl-Gal	C_24_H_22_O_14_	534.1014	534.101	557.0907	+	-	-	+	0.4	[25]
**29**	2.188	Kaempferol-3-*O*-acetyl Glc	C_23_H_22_O_12_	490.118	490.1111	491.1186	+	-	-	+	0.7	[40]
**30**	4.918	Quercetin	C_15_H_10_O_7_	302.0432	302.0427	303.0498	+	-	-	+	1.5	standard

E—ethanol extract; F1—water fraction; F2—40% methanol/water fraction; F3—70% methanol/water fraction; * Rut: rutinoside, Glc: glucoside, Gal: galactoside, and Ara: arabinoside; and ** XlogP3: according to PubChem data base.

**Table 3 molecules-28-07459-t003:** Content of phenolic compounds in the blackcurrant leaf ethanol extract and its fractions as determined by HPLC (mg/g extract).

No	R_t_ [min]	Compound *	E	F1	F2	F3
**9**	9.55	Gallic acid	2.821 ± 0.011	1.74 ± 0.02	n.d.	0.332 ± 0.011
**6**	10.20	Chlorogenic acid	2.85 ± 0.002	0.12 ± 0.01	n.d.	0.171 ± 0.001
**7**	11.10	Neochlorogenic acid	1.364 ± 0.002	0.025 ± 0.013	0.738 ± 0.011	n.d.
**18**	16.45	*p*-Coumaric acid	0.433 ± 0.011	n.d.	1.800 ± 0.009	0.594 ± 0.011
**13**	17.51	Quercetin-3-*O*-Rut	0.386 ± 0.004	n.d.	0.254 ± 0.003	n.d.
**20**	17.99	Ferulic acid	1.637 ± 0.017	n.d.	24.663 ± 0.012	n.d.
**22**	18.28	Kaempferol 7-*O*-Glc	0.029 ± 0.019	n.d.	1.359 ± 0.002	0.170 ± 0.012
**15**	18.39	Quercetin-3-*O*-Glc	2.303 ± 0.014	n.d.	68.499 ± 0.022	0.285 ± 0.017
**16**	19.17	Quercetin-3-*O*-Gal	1.090 ± 0.008	n.d.	27.112 ± 0.023	0.382 ± 0.012
**19**	21.01	Quercetin 3-*O*-malonyl-Glc	2.085 ± 0.006	0.121 ± 0.014	0.152 ± 0.012	41.852 ± 0.015
**11**	22.53	Rosmarinic acid	0.254 ± 0.012	n.d.	15.336 ± 0.012	n.d.
**12**	33.19	Quercetin	0.692 ± 0.021	n.d.	n.d.	7.92 ± 0.052
		Total	15.944	2.006	139.913	51.706

E—ethanol extract; F1—water fraction, F2—40% methanol/water fraction; and F3—70% methanol/water fraction. * Rut: rutinoside, Glc: glucoside, and Gal: galactoside, n.d.—not detected.

## Data Availability

The data obtained for the publication of this article are available upon reasonable request to the corresponding author.

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
