# Peer review of "Phenolic Compounds in Fractionated Blackcurrant Leaf Extracts in Relation to the Biological Activity of the Extracts"

_molecules, 2023, doi:10.3390/molecules28227459_

Round 1

Reviewer 1 Report

Comments and Suggestions for Authors

The present work by Staszowska-Karkut et al. entitled, ‘Phenolic compounds in fractionated blackcurrant leaf extracts in relation to the biological activity of the extracts’ focuses on the relationship between phenolic components from ethanolic extract of blackcurrant leaves and its fraction vis-à-vis their antioxidant and anticancer activities. The authors extracted blackcurrant leaves, and then fractionated the extract into three portions via SPE with water/methanol of different solvent ratios. The TPC and TFC were determined as well as their in vitro antioxidant and anticancer properties against two cancer cell lines. The relationship between key phenolic compounds identified and the observed biological attributes was established. The work presents an interesting perspective on the rich phenolic profile and biological properties of both blackcurrant leave ethanolic extract as well as its main bioactive fraction (F2), highlighting their potential as source of valuable active ingredients for pharmaceutical or functional food application.  Indeed, the authors deserve commendation of the clarity in which they presented their work and major findings. However, there are some questions which are yet to be addressed and it would be worthwhile for the authors to consider them.

-On the choice of fractionation solvent, why use methanol even though it is known to be toxic? And even after evaporation, there is always the risk of residual solvent in the sample.

-In the text, the IC50 values are provided in mg/mL whereas in Table 2 they are presented in µg/mL. Which of these is correct?

-Details on how the IC50 values were determined should be included in the methods.

-Please provide the ATCC number of the cell lines used in this work.

-The reference protocol used for determination of anticancer properties, ‘EN ISO 10993-5:2009’, is actually intended for material devices and extracts prepared from such rather than for plant samples or natural products. How do the authors justify using a protocol that is clearly meant for a different purpose for another even when there are well-established in vitro protocols for screening or determining the cytotoxicity of natural products such as that of the US National Cancer Institute

-Based on the authors’ findings, what IC50 value would be considered non-cytotoxic against the cancer cell lines for the extract as well as fractions?

-Is the extract and/or F2 selective against non-cancer cells, and if so, to what extent? It would be valuable to support this with some data.

-The investigation is completely lacking in terms of mechanistic insight responsible for the anticancer properties of the extract and F2. Authors should consider providing some data to support directly or indirectly the anticancer mechanism of the active components. This gap should be addressed.

-The metabolite profile of F2 and F3 are almost similar, yet only F2 exerted activity against the colorectal cancer cell line. Is there any reason for this observation?

Please recheck the ‘total’ values in Table 4, some of them did not add up.

The LC-MS methodology, authors should indicate the name and manufacturer of the column used in the LC compartment.

-Figure 2 image quality is poor. It should be improved.

Comments on the Quality of English Language

The entire manuscript should be thoroughly edited to remove minor errors in the writeup.

Author Response

Thank you very much all Reviewers for your comments on our manuscript: molecules-2682526; Title: Phenolic Compounds in Fractionated Blackcurrant Leaf Extracts in Relation to the Biological Activity of the Extracts

Responses to reviewers comments and information regarding corrections made to the manuscript are listed below.

Reviewer #1: 

  1. On the choice of fractionation solvent, why use methanol even though it is known to be toxic? And even after evaporation, there is always the risk of residual solvent in the sample.

Answer: Methanol was used in the experiment for economic reasons, it is much cheaper than ethanol. The conducted research was at the in vitro level. We assumed that freeze-drying would allow complete removal of methanol from the tested samples because a very low vacuum was used, at which it is even possible to remove adsorption-bound water (0.008 mBar). Thank you for this valuable comment, which we will take into account when planning in vivo studies

  1. In the text, the IC50 values are provided in mg/mL whereas in Table 2 they are presented in µg/mL. Which of these is correct?

Answer:  The correct IC50 is µg/mL. Appropriate corrections have been made to the manuscript (L. 156, 160).

  1. Details on how the IC50 values were determined should be included in the methods.

Answer: The information has been added: Dose-response curves were generated by graphing the cell growth percentage against the logarithmic concentration of the compound. The IC50 (concentration that inhibits 50%) values were determined through a non-linear regression model applied to the sigmoidal dose-response curve (L. 584-587)

  1. Please provide the ATCC number of the cell lines used in this work.

AnswerHCT 116 (ATCC® CCL-247), PC-3 (ATCC® CRL-1435), L929 (ATCC® CCL-1)

  1. The reference protocol used for determination of anticancer properties, ‘EN ISO 10993-5:2009’, is actually intended for material devices and extracts prepared from such rather than for plant samples or natural products. How do the authors justify using a protocol that is clearly meant for a different purpose for another even when there are well-established in vitro protocols for screening or determining the cytotoxicity of natural products such as that of the US National Cancer Institute

Answer: The ISO standards referenced in our work do not specifically address natural extract-based preparations. However, they encompass general guidelines for good cell culture practices, provide recommendations for analytical methods, and endorse the utilization of the L929 cell line, which we employed in our cytotoxicity studies. The overarching principles outlined in ISO are adhered to within our laboratory, where we conduct extensive assessments on a wide range of substances and materials, including those related to nanotechnology.

  1. Based on the authors’ findings, what IC50value would be considered non-cytotoxic against the cancer cell lines for the extract as well as fractions?

Answer: The manuscript has been supplemented with the information suggested in the review (L. 176-184). 

  1. Is the extract and/or F2 selective against non-cancer cells, and if so, to what extent? It would be valuable to support this with some data.

 Answer: Cytotoxicity tests were performed on L929 mouse fibroblasts, which are used as normal cells in this type of research. No other studies on the cytotoxicity of the tested samples with respect to normal cells have been conducted so far.

  1. The investigation is completely lacking in terms of mechanistic insight responsible for the anticancer properties of the extract and F2. Authors should consider providing some data to support directly or indirectly the anticancer mechanism of the active components. This gap should be addressed.

Answer: The manuscript was supplemented with information on the possible mechanisms of action of phenolic compounds in the context of their anti-cancer activity (L.326-385). Thank you for your comment.

  1. The metabolite profile of F2 and F3 are almost similar, yet only F2 exerted activity against the colorectal cancer cell line. Is there any reason for this observation?

Answer: Fraction F2 was the richest in phenolic compounds, which was confirmed by spectrophotometric (TP) and HPLC analyses. Statistical analyzes allowed to determine the relationships between the determined compounds and the anticancer activity of extracts and fractions. The obtained results confirmed the influence of some of the tested compounds on the activity of HCT 116, which dominated in the F2 fraction.

  1. Please recheck the ‘total’ values in Table 4, some of them did not add up.

Answer: Thank you for your comment. Appropriate correction has been made to the manuscript (L. 316)

  1. The LC-MS methodology, authors should indicate the name and manufacturer of the column used in the LC compartment.

Answer: The LC-MS methodology was supplemented with the name and manufacturer of the column used for analyses (L. 595)

  1. Figure 2 image quality is poor. It should be improved.

Answer: The quality of Figure 2 has been improved.

Additional changes made in manuscript:

  1. Table 2 and 3: Full names of phenolic derivatives were changed on shorter form.
  2. Spearman's correlation analysis was done and results were inserted in Supplementary Table S-2

According to  the reviewers' suggestions, the manuscript was linguistically revised again.

All changes made in the manuscript were marked in red.

Reviewer 2 Report

Comments and Suggestions for Authors

The authors of the manuscript entitled "Phenolic Compounds in Fractionated Blackcurrant Leaf Extracts in Relation to the Biological Activity of the Extracts" describe the phytochemical, antioxidant, and anti-cancer properties of blackcurrant leaf extract and three fractions.

Up-to-date literature abounds with similar investigations on the other plant materials rich in phenolics and the results of this manuscript are rather predictable, however, the presented manuscript along with confirmation of some previous findings complements the literature and adds new knowledge on the pharmacological importance of blackcurrant extracts and their phenolic compounds.

There are some issues that need to be addressed before reconsideration:

1.  The scientific description of the plant (blackcurrant), including its Latin name, is missing.

2.  Line 373: In what solvent were samples redissolved prior to phytochemical, antioxidant, and anticancer analysis?

3.  Lines 103, 244: Please clarify the meaning of “chemical activity”.

4. Division of the Results and Discussion section into more subsections would ensure a well-structured manner of the manuscript. Also, there are two 2.2. subsections (Lines 182, 333), as well as two 3.2.2. subsections. Please check the numbering of the manuscript.

5. It would be easier to interpret and understand data if they were visualized in a chart rather than a table (e.g. Tables 1 and 2)

6. It would be interesting to compare IC50 values against cancer cell lines (not necessarily the same) of your obtained fractions to phenolic fractions from other plant materials reported previously.

7. It would be beneficial to discuss the possible realistic application of particular phenolic compounds from blackcurrants in the chemotherapeutic strategies of prostate or colorectal cancer. How these compounds are metabolized, do the metabolites are active or do these compounds express their effects only in an unaltered state? Can these compounds be detected in plasma after ingestion? How could they reach target sites – colon or prostate?

8.  A better description of PCA analysis and its outcomes (description of Figure 2) is needed (location of all samples in relation to all variables).

9. Why solid phase extraction as fractionation technology was selected? What is the yield of such a method (how much of fractions in mg were obtained from X g crude extract)?

10. Other than phenolic compounds presence in crude extract and obtained fractions should be taken into consideration and discussed.

11. Did you measure the residual moisture in the extract and fractions? Were the extracts and fractions completely dry powders? Did you somehow remove chlorophyll from the leaves?

12.  I would suggest the authors build up the conclusion section towards pointing out the most important findings of this study and critically pointing to the particular perspectives of further research. If you recommend blackcurrants in health promotion, what markers should be used for quality control of blackcurrants functional food and dietary supplements?

Comments on the Quality of English Language

A further focus on concise and clear English language in writing would improve the manuscript significantly.

Author Response

Thank you very much all Reviewers for your comments on our manuscript: molecules-2682526; Title: Phenolic Compounds in Fractionated Blackcurrant Leaf Extracts in Relation to the Biological Activity of the Extracts

Responses to reviewers comments and information regarding corrections made to the manuscript are listed below.

Reviewer #2: 

  1. The scientific description of the plant (blackcurrant), including its Latin name, is missing.

Answer: The manuscript was supplemented with the Latin names of black currant. Thank you for this remark (L. 15, 28, 47, 504).

  1. Line 373: In what solvent were samples redissolved prior to phytochemical, antioxidant, and anticancer analysis?

Answer:  Samples intended for microbiological analyzes were dissolved in DMSO. the remaining samples were dissolved in 80% ethanol, water, 40% and 70% methanol, respectively. The methodology in the manuscript has been extended with this information (L. 518-521).

  1. Lines 103, 244: Please clarify the meaning of “chemical activity”.

Answer: Chemical activity” has been used in the context of anti-radical and anti-cancer activities. However, this mental shortcut is not entirely accurate, so it was corrected as: „anti-radical and anti-cancer activity” in manuscript. (L. 108, 266)

  1. Division of the Results and Discussion section into more subsections would ensure a well-structured manner of the manuscript. Also, there are two 2.2. subsections (Lines 182, 333), as well as two 3.2.2. subsections. Please check the numbering of the manuscript.

Answer: The manuscript numbering was checked and appropriate corrections were made. Additionally, in the Results and Discussion, there is a separate subchapter "Biological properties"(L. 148).

  1. It would be easier to interpret and understand data if they were visualized in a chart rather than a table (e.g. Tables 1 and 2)

Answer: According to the reviewer's suggestion, Table 2 was replaced with a figure. Table 1 was unchanged because it presents the results expressed in different units.

  1. It would be interesting to compare IC50values against cancer cell lines (not necessarily the same) of your obtained fractions to phenolic fractions from other plant materials reported previously.

Answer: Information on this topic has been added to the manuscript in relation to the PC-3 line. (L. 196-200)

  1. It would be beneficial to discuss the possible realistic application of particular phenolic compounds from blackcurrants in the chemotherapeutic strategies of prostate or colorectal cancer. How these compounds are metabolized, do the metabolites are active or do these compounds express their effects only in an unaltered state? Can these compounds be detected in plasma after ingestion? How could they reach target sites – colon or prostate?

Answer: Thank you for this remark. The above-mentioned suggestion goes much beyond the scope of the presented work. Despite this some of these issues are included in the discussion (L. 422-450)

  1.  A better description of PCA analysis and its outcomes (description of Figure 2) is needed (location of all samples in relation to all variables).

Answer: The description of the PCA statistical analysis has been extended and supplemented (L. 467-479, 481-484)

  1. Why solid phase extraction as fractionation technology was selected? What is the yield of such a method (how much of fractions in mg were obtained from X g crude extract)?

Answer: The solid phase extraction (SPE) is one of the simplest and universal method of sample concentration and fractionation. Additionally silica gel modified with octadecyl (C18) is the most popular sorbent used for the isolation of phenolic compounds. (Perez-Magarin, 2008). As a result of this extraction, 368.5 mg of aqueous fraction, 212.4 mg of fraction in 40% methanol and 39.1 mg of fraction in 70% methanol were obtained, per 1 g of extract. This information was included in the manuscript (L. 98-101).

  1. Other than phenolic compounds presence in crude extract and obtained fractions should be taken into consideration and discussed.

Answer: Thank you for this remark. An appropriate comment has been added to the discussion of the results. (L. 310-314)

  1. Did you measure the residual moisture in the extract and fractions? Were the extracts and fractions completely dry powders? Did you somehow remove chlorophyll from the leaves?

Answer: Thank you for this valuable comment. The residual moisture was not measured in the extract after lyophilization. The extract and fractions were freeze-dried to obtain completely dry powders. Freeze-drying of samples allowed for almost complete removal of water from the tested plant material because a very low vacuum was used, at which it is even possible to remove adsorption-bound water (0.008 mBar). Chlorophyll was removed from the sample during the solid phase extraction process. Chlorophyll showed greater affinity for the C18 sorbent than for the solvents used to elute phenolic compounds (water, 40%MeOH, 70%MeOH).

  1. I would suggest the authors build up the conclusion section towards pointing out the most important findings of this study and critically pointing to the particular perspectives of further research. If you recommend blackcurrants in health promotion, what markers should be used for quality control of blackcurrants functional food and dietary supplements?

Answer: Thank you for this remark. As suggested by the reviewer, the conclusions indicate the main findings and highlight research problems that must be solved before using blackcurrant leaf extracts in food, cosmetics and medicine. (L. 646-650, 653-655)

Perez-Magarino, S.; Ortega-Heras, M.; Cano-Mozo, E. Optimization of a Solid-Phase Extraction Method Using Copolymer Sorbents for Isolation of Phenolic Compounds in Red Wines and Quantification by HPLC. J. Agric. Food Chem. 2008, 56, 11560–11570.

Additional changes made in manuscript:

  1. Table 2 and 3: Full names of phenolic derivatives were changed on shorter form.
  2. Spearman's correlation analysis was done and results were inserted in Supplementary Table S-2

According to  the reviewers' suggestions, the manuscript was linguistically revised again.

All changes made in the manuscript were marked in red.

Round 2

Reviewer 2 Report

Comments and Suggestions for Authors

The authors have addressed all the raised questions and concerns and the manuscript improved significantly after revision.

Line 200. Please check the spelling.